**communications** engineering

# Meso-scale seabed quantification with geoacoustic inversion
Tim Sonnemann [1,2] ✉, Jan Dettmer [1], Charles W. Holland [2] & Stan E. Dosso [3]

Knowledge of sub-seabed geoacoustic properties, for example depth dependent sound speed and porosity, is of importance for a variety of applications. Here, we present a semi-automated geoacoustic inversion method for autonomous underwater vehicle data that objectively adapts model inference to seabed structure. Through parallelized trans-dimensional Bayesian inference, we infer seabed properties along a 12 km survey track on the scale of about 10 cm and 50 m in the vertical and horizontal, respectively. The inferred seabed properties include sound speed, attenuation, density, and porosity as a function of depth from acoustic reflection coefficient data. Parameter uncertainties are quantified, and the seabed properties agree closely with core samples at two control points and the layering structure with an independent sub-bottom seismic survey. Recovering high resolution seabed properties over large areas is shown to be feasible, which could become an important tool for marine industries, navies and oceanic research organizations.

Knowledge about the composition of the seabed below the water-sediment interface is of critical importance for the well-being of many nations. For example, populations increasingly rely on infrastructure on continental shelves for renewable energy generation[1–3] and trans-oceanic communication links[4] that require seabed geohazard evaluations[5] and environmental assessments[6]. Navies require seabed knowledge for many sonar applications such as anti-submarine warfare[7], sonar performance modeling, and mine burial detection[8]. The study of marine animal populations and behavior also benefits from such knowledge[9]. These applications will benefit from improved lateral and vertical seabed resolution. Instruments and methods that survey the seabed as a function of depth are limited to direct sampling (e.g., coring[10]) or employing underwater sound waves as in high-resolution chirp sonar, marine seismic reflection, and refraction surveys[11]. None of these approaches have provided to date both sufficient resolution and material properties required for next-generation survey applications. Seabed studies remain a research frontier due to the high cost of operating research vessels and due to the vast areas with little existing knowledge.

Remote sensing with autonomous platforms such as satellites and drones has profoundly changed our ability to estimate topography and surface properties on land. In the oceans, autonomous underwater vehicles (AUVs) are utilized increasingly due to their efficiency and broad applicability to many scientific, commercial and military areas[12–14]. AUVs are used in oceanography to acquire data[15,16]; in the oil and gas industry as well as off-shore wind energy projects to survey the seabed and monitor seafloor installations[17,18]; for port and harbor security tasks such as

undersea surveillance[19], search and rescue efforts[20], and detecting undersea mines[21].

Seabed properties of interest, such as porosity, density, and compressional-wave (sound) speed and attenuation, can be estimated by exploiting acoustic waves that interact with the seabed. We term these properties, geoacoustic properties in the remainder of this paper. Since geoacoustic properties are influenced by a wide variety of geological and biological processes that operate on a vast range of temporal and spatial scales, high spatial variability is common but largely unmapped. Over the past six decades, the principal observations used to estimate geoacoustic seabed properties have involved acoustic propagation over horizontal spatial scales of $O(10^3)$ to $O(10^4)$ m. Most often, the seabed is assumed as laterally invariant, or range independent, over such scales. However, it is clear from modeling that ignoring this dependence can lead to biases in the property estimates[22–24]. At much finer scales, direct sampling methods, such as cores and in situ probes, provide understanding of vertical variability at $O(10^{-2})$ to $O(10^0)$ m. But direct sampling is costly due to the need of ship time, has limited penetration depth, and the sampling process can disturb the sample. Comparatively little is understood about horizontal variability at $O(10^0)$ to $O(10^3)$ m, which we term geoacoustic meso-scale variability [25]. Studying meso-scale geoacoustic properties with existing methods is cost prohibitive over large areas.

We present a method for meso-scale geoacoustic seabed quantification (MGSQ) through automated analysis of survey data that adapts a seabed model to the structure resolved by the data. The method greatly simplifies

[1]Department of Geoscience, University of Calgary, Calgary, AB T2N 4N1, Canada. [2]Department of Electrical and Computer Engineering, Portland State University, Portland, OR 97201, USA. [3]School of Earth and Ocean Sciences, University of Victoria, Victoria, BC V8P 5C2, Canada. ✉e-mail: tim@pdx.edu

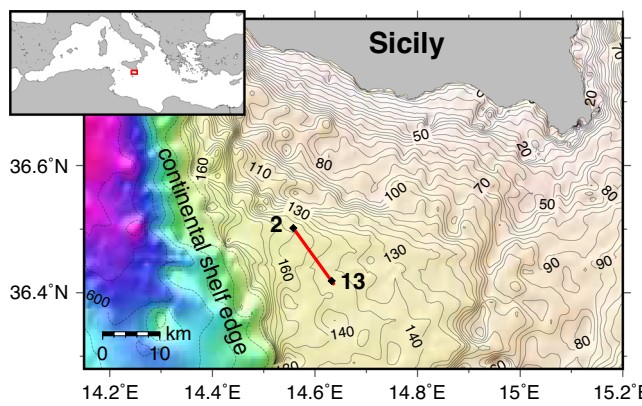

**Fig. 1 | Map of the study area off the coast of Sicily.** The survey track (red line), bathymetry (black contours, solid every 5 m until 195 m, dashed every 100 m from 200 m downwards), and core locations at sites 2 and 13 (black diamonds) are shown.

seabed surveys by reducing time, cost, and subjective operator choices in producing seabed images. We collected acoustic data along a 12 km track on the Malta Plateau in the Mediterranean Sea (Fig. 1) using an AUV towing an impulsive acoustic source and a linear array of 32 hydrophones (Fig. 2a). Even though an AUV was employed here, the method is equally applicable to towed sources and arrays that can record direct and bottom-reflected arrivals for a large number of source transmissions (pings) along the survey track. Care must be taken with respect to position of source and array in the water column to ensure that direct and bottom-reflected paths can be separated. This is done by keeping the source and receiver sufficiently far from the seafloor. The recorded pings are arranged to correspond to common-depth-point (CDP) gathers. The specific experiment we employ permits processing of seabed reflection-coefficient data between 32 and 67 degrees and between 900 and 3400 Hz. We will demonstrate that these data can provide detailed knowledge about geoacoustic properties which is contained in the Bragg interference pattern of the reflection coefficient as a function of angle and frequency. The data inverted for the seabed model consist of reflection coefficient (RC) spectra as a function of seabed grazing angle (hereafter referred to as angle), averaged over 11 consecutive CDP gathers to reduce noise, resulting in 1711 distinct RC data sets. These RC data sets are inverted independently for geophysical parameters using a sediment acoustic model. The model used here is based on the grain shearing (GS) model[26], which considers unconsolidated sediments as an assemblage of mineral grains with their surfaces in contact and seawater filling interstices. Note that the GS model satisfies causality which leads to frequency dependence of attenuation and compressional wave velocity. GS model parameters include porosity, grain-to-grain compressional modulus, and material index (defines strain hardening). The GS parameters can be transformed to sediment parameters of bulk density, sound speed, and sound attenuation (the latter two are frequency dependent)[26], which are needed for the RC model. For each RC data set, a local 1D horizontally layered model is assumed (Fig. 2b), and model geoacoustic parameters and uncertainties are estimated which fit the observed data through trans-dimensional (trans-D) inference. The reversible jump Markov chain Monte Carlo algorithm is implemented for parallel execution on central and graphical processing units (CPUs and GPUs, respectively) using an efficient parallel tempering scheme and principal-component parameter space perturbations. Finally, the 1D inversion results are concatenated to a subsequently smoothed 2D geoacoustic model.

The result of the Bayesian inference are posterior probability densities (PPDs) of geoacoustic parameters at ten centimeter resolution in the vertical direction and tens of meters resolution in the horizontal. This previously unavailable resolution is important for applications such as identifying and mapping geohazards, quantifying benthic habitats, and understanding propagation of acoustic signals in the ocean. Standard acoustical methods

suitable for large areas (e.g., vertical-incidence seismic surveys) infer sound speed and a distorted depth estimate (based on two-way travel time), but rarely provide density, attenuation or porosity. We use the frequency-domain spherical-wave RC[27] and improved data processing methods to estimate those geoacoustic properties at high resolution for a large dataset within a reasonable time. That means MGSQ at scale is becoming computationally feasible.

## Results
### Meso-scale features can be resolved
The estimated depth- and range-dependent seabed properties are shown in Fig. 3a–d in terms of median values of the PPD for porosity, sound speed, density, and attenuation from inversion of each of the 1711 RC data sets along the track. These results show that we can resolve meso-scale lateral (range-dependent) features using reflectivity data obtained with an AUV, which is the main achievement of this study. Another important observation is that while the layering structure changes along the track, we are able to resolve this variation sufficiently well with the 1D assumption inherent in each of the individual inversions. The geoacoustic structure is inferred down to 6.8 m below the seafloor (defined by time-windowing of the acoustic data). The inversion also provided quantitative uncertainty estimation: Supplementary Fig. 2 shows 95% credibility interval widths for these results. The trans-D inversion is an appropriate tool to estimate such cases where both the number of layers and their parameter values are uncertain and change along the track. Due to variability between consecutive 1D inversions, which is considered to stem from less-well converged parameter chains, a horizontal 5-CDP-average median filter was applied to the along-track medium parameter estimates for visual clarity. The CDP averaging and subsequent smoothing results in a horizontal resolution of ~ 50 m, as the seafloor footprint of each CDP is 12 m, the distance between each CDP is 3.5 m, and we averaged 11 adjacent CDPs for each RC data set that was inverted.

### Porosity reveals mud and sand structure
The results include both the estimated GS parameters (discussed first) and the derived geoacoustic properties. The porosity and interface depth estimates allow detailed identification of sediment structure and type. The track results in Fig. 3a show estimated porosity values ranging from 0.3 to nearly 0.9, which likely correspond to sediments such as compacted sand and very soft mud, respectively. From top to bottom, the track shows a roughly 15 cm thick layer of about ~ 0.8 porosity which is interpreted as a mud. A relatively large decrease in porosity to about ~ 0.65 is observed next. This second layer decreases in thickness from north to south (left to right in Fig. 3) with an initial thickness of 1.2 m that reduces to less than 0.2 m and possibly even disappears. These upper two layers are muddy sediments, as indicated by their properties (and confirmed by core data, see Fig. 10 in[28]) and will be referred to as the mud wedge. Below that, a 1.6 and 3.2 m thick formation with ~ 0.55 porosity is observed. Core data indicate this to be mud mixed with considerable amounts of sand, shells and shell fragments with volume fractions that vary with depth. The inversion captures this as multiple layers with very similar parameter values. Small differences are resolved as weakly banded appearance. This is followed by an erosional unconformity which is characterized by large-scale interface roughness above a denser sediment layer of ~ 0.5 porosity, which is interpreted as moderately-compacted sand.

Structures deeper than the erosional unconformity have higher uncertainties. The GS parameters of the half-space (deepest layer) are poorly resolved and exhibit high porosity and high lateral variability. The high lateral variability (and possibly other characteristics) is not interpreted to be representative of the actual seabed, but rather is an artifact of limited information content of the data for the half-space. The principal information content in RC data comes from Bragg interference within a layer. A half-space, by definition, produces no Bragg interference, hence the only information is a weakly-varying angular dependence of reflections off the half-space boundary.

**Fig. 2 | Data acquisition setup with an autonomous underwater vehicle (AUV). a** AUV, source and hydrophone array on the surface. Photo courtesy of Christopher H. Harrison. **b** Schematic of autonomous data acquisition system (components not to scale). The AUV tows an acoustic source and array of hydrophones close to the seabed and records direct arrivals and bottom and sub-bottom reflections illustrated by ray paths.

**Fig. 3 | Median of posterior probability densities of estimated geoacoustic parameters. a** Porosity, (**b**) sound speed $c_p$, (**c**) density $\rho$, and (**d**) attenuation $\alpha_p$. The track bathymetry is shown with depth relative to the sea surface. The layered structure is clearly visible.

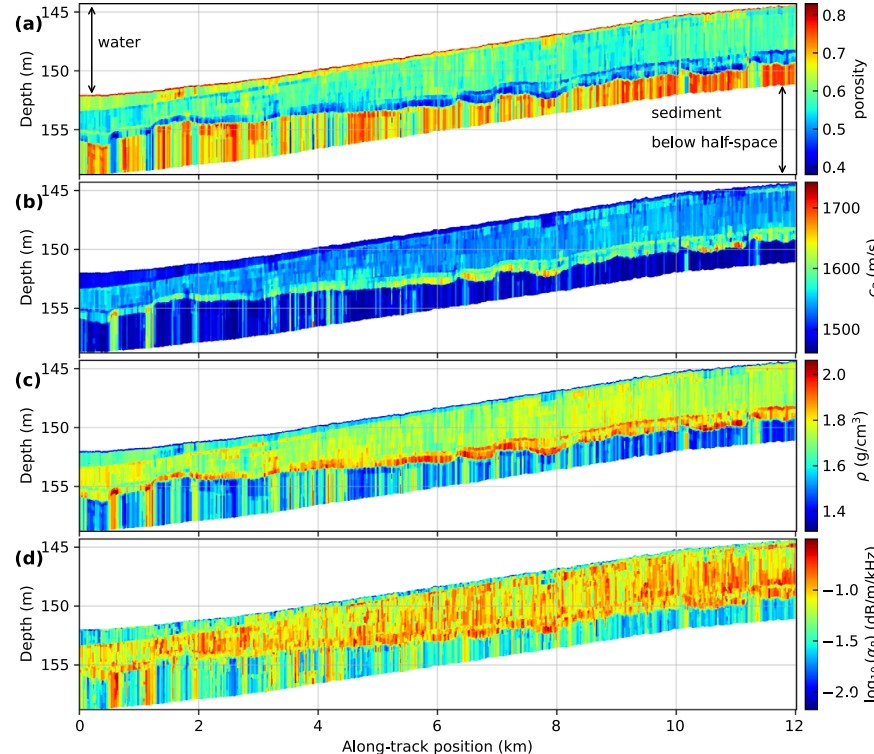

## Depth-dependent physical properties revealed

As shown above, the physical properties of the sub-bottom can be obtained with high vertical resolution. This knowledge can be used to identify sediment types and model the mechanical response of the sediments. Given uncertainties and potential artifacts, there appear to be no major lateral changes of the geoacoustic properties within the same layer despite varying layer thicknesses. This is an important observation that could inform geoacoustic scenario modeling efforts. Overall, sound speed (Fig. 3b) varies from 1450 to 1650 m/s and density (Fig. 3c) varies from 1.25 to 2.0 g/cm³, which are consistent with unconsolidated sediments. Sound speed and density consistently increase with decreasing porosity. The uppermost layer sound speed is about 1470 m/s, or a sediment-to-water sound speed ratio of 0.972 (given that the water sound speed was 1512.3 m/s), which is within the range of observed sound speed ratio measurements on fine-grained sediment cores[29]. Attenuation (Fig. 3d) is less well constrained, but the top two layers (the mud wedge) consistently exhibit a lower attenuation than the rest, each with a median of about 0.03 dB/m/kHz. The thick third and erosional fourth layers both have similar values about 0.1 dB/m/kHz. The compressional grain-to-grain modulus $\gamma_p$ is relatively low in the mud wedge and higher in the sediments below (Supplementary Fig. 3). This trend is expected inasmuch as cohesive sediments (mud) are expected to exhibit lower moduli than granular sediments[26]. The material index is smallest in the mud wedge, and somewhat poorly defined in the layers below it (Supplementary Fig. 3).

## Verification of sediment properties

Confidence in the sediment parameter estimates is established through comparisons to coring measurements, independent sub-bottom profiling, and data fit evaluations.

Independent measurements of sound speed and density were made on piston and gravity cores near the beginning and end of the track (at what are referred to as sites 2 and 13, respectively). A comparison of the inversion results (which were not informed by the cores) with the cores is shown in Fig. 4a, b. Core data have their own uncertainties in the sampling, recovery, and measurement phases, which unfortunately are difficult to quantify. Some discussion on the core data is given in[28]. Here, the individual inversion results closest to the core sites are shown as marginal probability profiles which indicate parameter uncertainty distribution as a function of depth. Both the layered model structure and the corresponding sound speed and density values are clearly different between the two sites, which is also indicated by the core data. The inversion results for sound speed and density agree well with the core measurements at both sites.

The observed RC data near sites 2 and 13 are well reproduced using the estimated model parameter values, as shown in Fig. 4c, d. It should be noted that the angle-to-angle variability of the observed data seems to be less than that expected due to random errors of the indicated standard deviations, which suggests overestimation of the variance.

In Fig. 5, the layer boundaries from the inversion (Fig. 5a) are compared to an independently acquired high-resolution seismic section

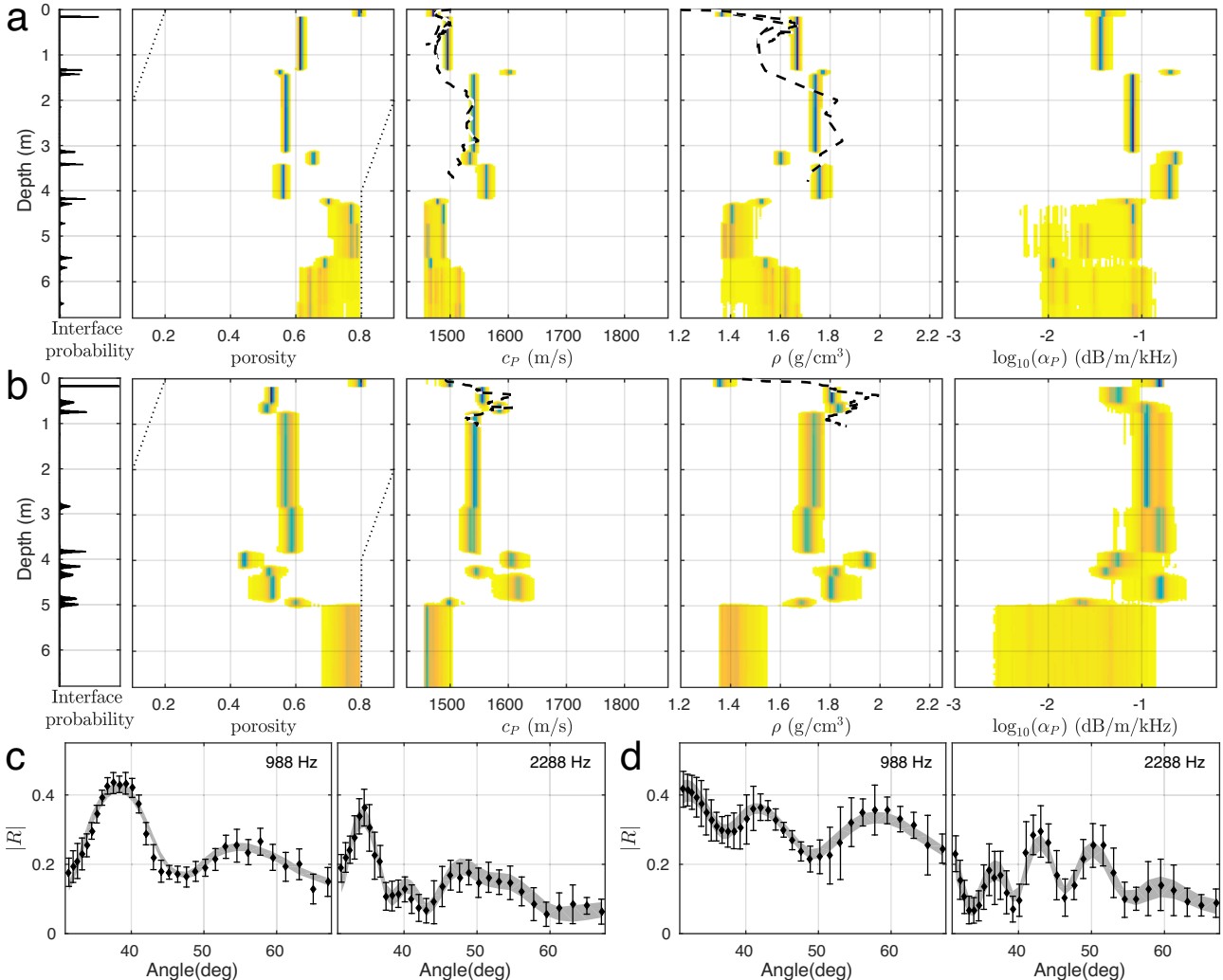

**Fig. 4 | Comparing estimated parameters to coring data and reflection coefficient data fit.** Inversion results for reflection coefficient data sets 82 (site 2: **a, c**) and 3338 (site 13: **b, d**). **a, b** Marginal probability profiles of estimated geoacoustic parameters. The compressional-wave speed and density have been measured at sites 2 and 13 by gravity and piston cores (dashed black-white lines), which agree well with the estimated values both in magnitude and depth. Prior probability bounds are indicated by dotted lines, profile densities are colored blue to yellow for high to low values, respectively, and white for zero density. **c, d** Examples of observed data (black diamonds with one standard error) and 95% credibility interval of predicted data (gray bands) which indicate a good model-to-data fit.

(Fig. 5b). These seismic data were collected close to the survey track using a surface towed boomer source[27] and short towed array. The seismic depth axis is distorted and is based on the assumption that sound speed is 1500 m/s in all sediment layers, while our RC inversion results indicate true depth estimates. The towed boomer exhibited some ringing which yields artificial (duplicate) parallel lines in the seismic section. Accounting for these differences, the porosity inversion results agree well with the seismic section, with all layer boundaries being represented and their shapes largely matching.

Distribution estimates of important statistics along the track are shown in Supplementary Figs. 1, 2, and are explained in Supplementary Note 1. Root-mean-square residuals, number of inferred layers (Supplementary Fig. 1), and credibility interval widths (Supplementary Fig. 2) are presented to support statements concerning spatial resolution and parameter estimate verification.

## Discussion

### High information content in frequency-angle space

It may seem surprising that such detailed geoacoustic information (vertically and laterally) could be inferred from such angle-limited data, $\sim 32 - 65°$. This is explained by the fact that substantial geoacoustic information content is contained in the Bragg interference structure, i.e., the interference pattern between the up- and down-going waves within each layer. This interference structure can be observed over a relatively modest angular/frequency range, and a broad frequency range can partially compensate for the modest angle range. An example of this is given in the Methods section. Angular diversity is required to break the ambiguity between thickness and sound speed in a given layer. The angular range in this data set is clearly sufficient to do so – note the small uncertainties in layer thickness (layer horizons) and sound speeds as indicated in Fig. 4 a and b. The steepest angle, 65°, and the total bandwidth, $\Delta f = 2700$ Hz, define the thinnest resolvable layer, which is $\sim c_i/(4\Delta f \max(\sin \theta_i))$ where $c_i$ and $\theta_i$ are the sound speed and grazing angle of the $i$-th layer, respectively. For example, for a layer sound speed of 1500 m/s, the thinnest resolvable layer is $\sim 0.15$ m.

In an ideal survey geometry, the seabed reflection angles would extend from below to above the critical angle. In the present data set, the angles are nearly all above the critical angle. Measuring more angles below the critical angle requires either a longer receive array or receivers closer to the seafloor. Small receiver heights, however, risk bottom entanglement and also degrade the data quality. That is, as the receiver height decreases, the Fresnel zone size decreases and hence the reflection data have larger errors from small-

**Fig. 5 | Comparing estimated sub-bottom structure to seismic reflection profile. a** Posterior probability density median of porosity with transparency scaled by 95% credibility interval width and core measurement sites 2 and 13 annotated. **b** Seismic reflection profile along autonomous underwater vehicle track. In the inversion result, the sediment layers are well resolved, especially the mud wedge along track which pinches out towards the center, and the erosional layer that is between 3 and 4 m below the sea bottom. Deeper layers are less certain, although still discernible.

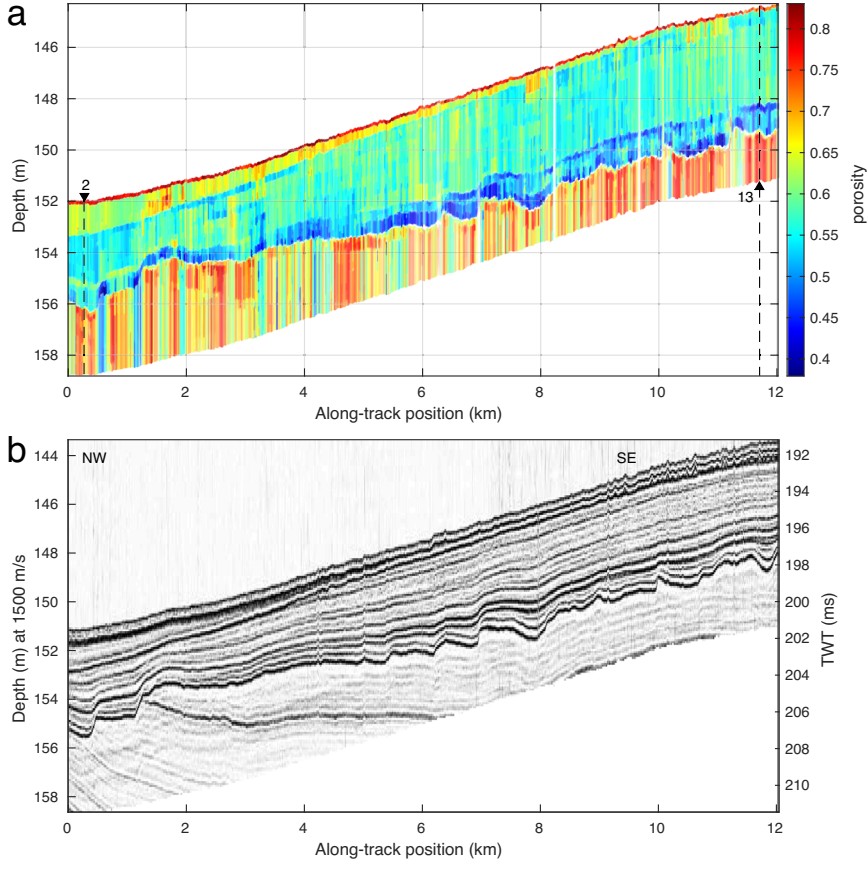

### Erosional layer related to last Ice age

The erosional unconformity which is shown between 3 and 4 m below the sea bottom in Figs. 3, 5 is most likely due to the sea level regression during the last glacial maximum (LGM), between 19 and 23 ka BP[30]. The relative sea level in the Mediterranean Sea was up to about 145 m lower during the LGM and the Malta Plateau was partially above sea level or barely submerged in the shallow surf zone and affected by eroding wave activity[31]. Radiometric dating of deeper cores obtained about 40 to 60 km north-west of our study area off the southern Sicilian coast indicates the unconformity's relationship to the LGM[32]. The resolved layers of this study likely correspond to unit 1 and the upper unit 2 identified in a seismic survey conducted in our area[33]. Unit 1 has been characterized as Holocene highstand shelf sediment consisting of shallow water calcareous muds with shells, shell fragments and coarse terrigenous material. Unit 1 is bounded by an erosional horizon previously detected with a thickness of 5–10 ms two way travel time (3.75–7.5 m at 1500 m/s). The upper boundary of unit 2 has carbonate buildup structures typical for a shallow water environment which suggests a very low sea level during its formation[33].

Typically, sub-bottom structure studies rely on two-way travel time of seismic reflection surveys without precise depth estimates. Here, the RC inversion method offers survey-wide absolute depth values for all structure. Another advantage is that knowing the physical material properties can aid considerably in geological interpretation. The present study has revealed high sound speed and density in isolated pockets in the swales of the erosional layer. These could be occurrences of coarse lag sediments that would tend to be deposited there; alternatively, they may be artifacts due to acoustic focussing effects (discussed later).

### Implications for ocean acoustic propagation and reverberation

Long-range acoustic propagation and reverberation are important for a wide variety of scientific and military purposes. Geoacoustic properties can profoundly affect both quantities. For example, it was shown theoretically (and confirmed numerically) that a mud wedge leads to surprisingly large transmission losses[24]. This is a general result, i.e., holds for arbitrary mud thickness profiles. The mud thickness profile in this environment would most dramatically increase transmission losses at mid-frequencies of 1–10 kHz. Long-range acoustic reverberation is governed by steeper angles than transmission loss (since scattering strength increases with increasing angle) and the geoacoustic properties here indicate that mid-frequency reverberation would not be controlled by the water-sediment interface, which has no critical angle and a relatively weak density contrast. Rather, reverberation would be controlled by either sediment volume heterogeneities and/or interface roughness of the erosional unconformity. Clutter, i.e., sharp peaks in the reverberation, is generally thought of as arising from discrete features. However, it was shown that clutter can arise from a slowly-varying seabed environment, where there is a surficial mud layer with varying thickness[34]. The variable mud thickness leads to specific points in the seabed where a resonant condition is met resulting in a high incident field at the base of the mud which in turn leads to high reverberation peaks or clutter returns at that location. It was also shown that clutter can arise from lateral sound speed gradients in the mud as weak as 0.07 s$^{-1}$ even when the mud thickness is constant. In this environment, these clutter mechanisms are most likely to occur at mid-frequencies.

### Limitations regarding local 1D model assumption

There are a few ostensibly strong lateral heterogeneities in the top 1 m of the transect, observed as very high porosity zones in the mud wedge, e.g., at 1.8, 2.0, 3.2, 7.9 km in Fig. 3a. While it is possible that some local sediment

volumes might be variably compacted or cemented, we believe that these anomalies are more likely due to the breakdown of our local 1D model assumptions. There can be focussing and defocusing effects due to the roughness of both the seafloor and interfaces of strong reflectors while lateral heterogeneities can cause scattering. The low porosity zone in the lower half-space at about 500–600 m along-track position in Fig. 3a may have been caused by the abrupt step in the overlaying unconformity. The high porosity zone in the top 1 m at about 7.8 to 8.0 km along-track could have been caused by wave focussing due to the concave shape of the strongly reflective unconformity below. These 2D effects have not been accounted for due to our 1D assumptions. 3D effects from unknown structure perpendicular to the survey line could also affect the reflectivity values. Further, the error model assumptions could distort some of the parameter estimates and uncertainties. However, we are confident that this pragmatic approach yields useful results overall.

### Bayesian MGSQ is becoming computationally feasible

Large data volumes and computationally-intensive forward modeling mean that computational costs need to be carefully considered. Inverting each RC data set separately, we were able to use multiple high performance clusters and fully utilized GPU nodes due to the efficient implementation[35,36]. Each inversion took 12 hours using one node of 32 or 40 CPUs with 4 Nvidia V100 GPUs, or 24 h using 32 CPUs with 2 Nvidia P100 GPUs. As the 1711 RC data sets were inverted in batches of 100 on multiple Compute Canada clusters without specially assigned project resources, the wait times to balance fair usage added to the overall processing duration. The full inversion therefore took about 2 months after the final setup and configuration were determined. However, if resources were expressly dedicated it could take a week or less. While this represents a relatively large computational effort in an academic environment, other fields such as hydrocarbon industry applications routinely use far greater resources. Without the massively parallel implementation and the Levin integration of the Sommerfeld integral for spherical RC modeling[35,36], this study would not have been feasible.

### Conclusions

Our results demonstrate the ability to survey sediment geoacoustic properties including porosity, sound speed, attenuation and density in a complex multi-layered seabed over scales of 10 km, with a resolution of about 10 cm in the vertical and 10s of meters in the horizontal dimension. In this study, the focus was on the upper 6 m of the seabed. This kind of quantitative sub-bottom survey capability has the potential to substantively advance a wide variety of marine scientific and commercial endeavors. Examples include geohazard assessments—needed for wind farms and other offshore structures, siting and burial of undersea communication and power cables. It is also expected to advance understanding of sound propagation in bottom-limited areas, such as continental shelves, which is important to marine research and policy development. Quantitative sub-seabed mapping, as opposed to current qualitative mapping is expected to help transform our understanding, and better use, of the ocean.

### Methods

#### Processing of data recorded by autonomous vehicle

Acoustic data were collected along a 12 km track on the outer shelf of the Malta Plateau, south of Sicily in the Mediterranean Sea (Fig. 1), as part of the Clutter 09 Experiment in May, 2009[25]. The water depth decreased approximately monotonically along the survey track from 152 m (site 2) to 145 m (site 13). The experiment is illustrated in Fig. 2b. The AUV traveled at a speed of 1.17 m/s and a height of about 12 m above the seabed. The AUV towed an acoustic source (an Ultra Electronics 2-100 MPS cylindrical projector mounted in a spheroidal-shaped tow body) and a horizontal array of 32 hydrophones. The distance between the AUV and the source was 2.6 m and the first hydrophone was 10.38 m from the source; the hydrophones were spaced at 1.05 m with a total array length of 32.55 m. The source consisted of two piezo-electric transducers generating linear

frequency modulated (LFM) pulses (pings) from 800 to 1400 Hz and 1600–3500 Hz. The LFM pulses have good noise rejection characteristics; by match-filtering the data, all noise is reduced that exhibits different spectral or temporal characteristics than the LFM[37]. At one ping every 3 s, the spatial ping interval is 3.5 m. The source-receiver data were arranged into CDP gathers, described in[25], resulting in a CDP width of 5 m, i.e. the seafloor reflection points of all source-receiver combinations of one CDP gather span 5 m. The Fresnel zone diameter on the seafloor at a frequency of 1000 Hz is 7.4 m, which means the seafloor footprint of a CDP combining the Fresnel zone and CDP width is about 12 m.

The acoustic data inverted here consist of seabed RCs over multiple grazing angles and multiple frequencies, computed as the ratio of spectral power of bottom-reflected to direct-path wave arrivals. The RC data processing, including correction for beam pattern and transmission path, and the CDP processing are as in[25]. However, the removal of an AUV-scattered acoustic return as described in that work is corrected and updated here.

There are several challenges in RC processing with a towed horizontal linear array. One is that, given the geometry (the receivers are all at endfire with respect to the source), an 'in-stride' source calibration is impossible except at endfire. The source beam pattern characteristics (as a function of frequency and vertical angle) were measured by towing the source over a vertical line array (VLA) of hydrophones. A more difficult problem was scattering from inside the AUV housing, which affected only the 1600–3500 Hz signal due to the source geometry. This AUV scattered return (ASR) arrives 4.8 ms after the direct path on each of the 32 receivers (source and receivers are essentially in a straight line). Furthermore, the ASR is temporally separable from the bottom reflected return for some but not all of the receivers. This part of the problem was addressed previously using coherent subtraction[25] to obtain the pressure time series with the ASR removed, $\hat{p}$, from the raw time series $p$. However, it was discovered that there was an error in the reflection data processing, which is clarified and rectified below. The data processing for the spherical wave reflection coefficient $|R_s|$ as a function of specular angle at the seabed $\theta$ was given in Eqs. (4) and (5) of [25], and is written here with a slight modification in notation as

$$|R_s(\theta, f, T; z_t)| = \frac{|\hat{P}_r(\theta, f, T; z_t)|}{|P_d(\theta_e, f, T_e)|} \frac{\gamma_d(\theta_e, f)}{\gamma_r(\theta, f)} \frac{|P_{dv}(\theta_e, f, T_e)|}{|P_{dv}(\theta, f, T_a)|} \frac{\gamma_{dv}(\theta, f)}{\gamma_{dv}(\theta_e, f)},$$

(1)

where $P$ and $\hat{P}$ are the Fourier transforms of $p$ and $\hat{p}$, respectively; $f$ is frequency; $z_t$ is the sum of the source and receiver heights; $\gamma$ is the transmission factor from source to receiver which includes spreading and absorption loss calculated via ray theory; subscripts d and r identify the path type: direct path and seabed reflected, respectively; $\theta_e$ is the grazing angle for the direct path to a receiver that is at endfire from the source; $T$ is the integration time with specific time windows $T_e$ and $T_a$ for the direct endfire path excluding the ASR and including the ASR, respectively; and subscript dv indicates a separate direct-path source beampattern measurement using the VLA. While there are several ways to process the time series data to compute $|R_s|$, this approach has the advantage of normalizing each term so that uncertainties in source, receiver, and data acquisition system calibrations play a negligible role. Also, although source depth and amplitude variabilities are modest, they are also removed through this approach.

In[25], the time window for all source to receiver direct paths was identical, $T_e = T_a$. This unintentionally assumed that the effect of the ASR on the bottom-reflected path was negligible. However, this effect was not negligible and led to artifacts in the reflection data. The processing used in this work corrects this by normalizing the bottom reflected path $\hat{P}_r$ (which inevitably contains contributions from the bottom reflected ASR) with the direct path calibration that also contains the ASR, $P_{dv}(T_a)$. This clearly reduced the artifacts in the data, which is important for the method generally. However, the specific impact of this processing on inversion results is not quantified in this study. Small artifacts remain because the normalization is at the specular

angle (the angle at the water-sediment interface) but paths that reflect from sub-bottom layers are steeper than specular.

Despite the data processing described above, an angle-dependent sawtooth pattern in computed RCs as a function of angle persisted at various frequencies throughout all CDPs along the entire track. An example is given in Supplementary Fig. 4a, which shows RC as a function of angle computed for an 11-CDP average centered on the 12th CDP gather, combining two frequency bands centered on 1100 and 1125 Hz. Note the oscillatory pattern above angles of 48° — similar patterns occurred across thousands of CDPs so were clearly not related to the environment. Excluding all angles and frequencies with this pattern causes too much data loss to resolve the seabed. Averaging RCs across angles smooths them out; however, that lowers resolution and conceals errors. As the artifact had only minor variability across all CDPs, a fifth-order polynomial was fit at each frequency to the mean of the RC data over all CDPs along the track (Supplementary Fig. 4c). The mean RC of the entire track included only very broad signal characteristics which were fit by the polynomials that otherwise passed through the finer sawtooth pattern. To obtain the corrected signal (Supplementary Fig. 4b), the difference between fitted polynomial values and the mean RCs of all CDPs (Supplementary Fig. 4d) was subtracted from the dataset. Removing the artifact in this way yields much clearer Bragg patterns in the data, produced by sediment layering.

After initial inversion trials for individual RC data sets failed to converge well (i.e, parameter values kept changing after many iterations) when using 4 frequency bands and all 32 grazing angles, simulations to estimate the required frequency content were carried out. Limited bandwidth and aperture were found to cause convergence issues with such data and geometries. Even a simple simulated model consisting of one layer over a half-space was found to pose difficulties when the angular range was too narrow. With a wider aperture that included the critical angle of reflection in the data, reliable inversions were obtained for simple models even with a small number of frequency bands. While extending aperture for the existing data is not possible, including additional frequencies was found to notably improve convergence, such that using eight frequency bands (centered at 988, 1113, 1263, 1913, 2288, 2513, 3013, 3313 Hz) was found to provide a well-determined model, despite the narrow aperture.

To reduce noise in the data, RCs for groups of 11 consecutive CDPs were averaged, which leads to an acoustic seafloor footprint of roughly 50 m. This lateral resolution corresponds well to our goal of investigating mesoscale variability without over-emphasizing smaller details. Inversion results presented here are obtained for data which averaged neighboring pairs of frequency bands for each CDP gather, the bandwidth for the inverted data was 50 Hz. No averaging across angles was carried out. To reduce the number of inversions, we used only every second of the 3431 available CDPs, which results in 1711 CDPs; considering the 11 CDP averaging means that the first and last CDP gathers are number 6 and 3426, respectively.

## Modeling response of unconsolidated sediments to sound waves

In any geoacoustic inverse problem, a sediment acoustics model must be chosen. However, in many works this choice is not discussed and most commonly a Hamilton model[38] is assumed with parameters of sound speed (without dispersion), attenuation (usually assumed to be a linear function of frequency), and density. Here, the sediment acoustics model employed is based on Buckingham's viscous grain shearing (VGS) model which, unlike the Hamilton model, obeys causality[26,39,40]. Causality is a useful constraint on parameter combinations; thus, one of the motivations of using the model is that it provides a fundamental limit on the search space. Further, it properly provides correlations between geoacoustic properties that should be correlated, e.g., bulk density and sound speed.

The specific implementation of the model is that described in[28], except that the viscoelastic time constant $\tau$ is set to infinity so that effects due to classical viscosity are ignored and the theory simplifies to the grain shearing (GS) model. We included $\tau$ in the initial stages of our investigation. However, multiple inversions indicated that $\tau$, material index, and $\gamma_p$ are strongly

correlated which caused problems with convergence of the inversion algorithm. To resolve $\tau$, a clear observation of classical viscosity in the sound speed dispersion and/or frequency-dependent attenuation is required. For example, classical viscosity leads to an attenuation that increases with frequency, $f$, as $f^2$ at low frequencies and $f^{\frac{1}{2}}$ at high frequencies. However, the data do not show such behavior. This could be either because classical viscosity plays a negligible role in these sediments over this frequency range, or there is an insufficient frequency and angular range to observe it. For either case, the inversion is best served by using the GS model. After applying the GS model to the inversion, reliable convergence was achieved. Also, as in[28], the grain-to-grain shear modulus $\gamma_s$ is not estimated directly from the data. The justification for this is that the conversion from compressional to shear waves is expected to be small since for unconsolidated sediments the shear speeds are low. Thus, the data appear to have insufficient information content to reasonably estimate the shear grain-to-grain modulus. Instead, $\gamma_s$ is inferred from the porosity $\beta$, which is highly sensitive to reflection data, via Eqs. (11), (12), (18) of Ref.[26] with constants $\gamma_{po}/\gamma_{so} = 10$, $\beta_o = 0.377$, $\beta_{min} = 0.37$ and $\Delta = 10^{-6}$. The bulk moduli and densities of grains and pore fluid have been fixed to these values: $K_s = 3.6 \cdot 10^{10}$ Pa, $K_f = 2.353 \cdot 10^{10}$ Pa, $\rho_s = 2.7$ g/cm³, and $\rho_f = 1.029$ g/cm³. In our implementation, three of the four GS parameters are inferred from the measured data, porosity $\beta$, the compressional grain-grain modulus $\gamma_p$ and the material index.

## Modeling reflection coefficient data

For each RC data set, we assume a 1D horizontally layered sediment model with an arbitrary number of homogeneous layers above a lower half space as shown in Fig. 2b. Due to the proximity of source and receiver to the seabed in the survey considered here, the plane-wave assumption is invalid and would lead to significant errors in modeling the observed RC values. Therefore, the spherical-wave RC is used instead, which is calculated by integrating plane-waves over all angles as expressed by the Sommerfeld integral

$$R_s(\theta, \omega) = G_\omega \int_0^\infty \frac{R_p(k_z, \omega)k_z}{k_r} e^{ik_r z_t} J_0(rk_z)\, dk_z, \tag{2}$$

where $\theta$ is the seabed grazing angle, $\omega$ is the angular frequency, $R_p$ is the plane-wave RC, $k_r = k\cos\theta$ and $k_z = k\sin\theta$ are the horizontal and vertical wavenumbers, respectively, $k = \omega/c_w$ is the wavenumber for water sound speed $c_w$, $z_t = 2H - (z - z_s)$, $H$ is the water depth, $z_s$ is the source depth, $z$ is the receiver depth, $J_0(\cdot)$ is the zeroth-order Bessel function of the first kind, $r$ is the horizontal range between source and receiver, $G_\omega = iDe^{-ikD}$, $D = \sqrt{r^2 + z_t^2}$, and $i$ is the imaginary unit[41,42]. The integrand in Eq. (2) depends on the seabed geoacoustic parameters only through the plane-wave reflection coefficient $R_p$, which can be computed by standard recursive algorithms[43]. The Bessel functions are known to be highly oscillatory, so the numerical evaluation of $R_s$ can be computationally expensive. In this work we use the efficient algorithm based on Levin integration and the hybrid CPU and GPU parallel implementation developed[35,36] to predict spherical RCs. The predicted RC data are frequency averaged similarly to the measured data.

## Bayesian inference

To obtain information on the model parameters including their uncertainty expressed by a potentially non-Gaussian posterior probability density (PPD), nonlinear Bayesian inference is used. This implementation considers the number of unknown parameters itself to be unknown, that is, the number of sediment layers is an unknown random variable in addition to each layer's physical parameters, which results in a trans-D Bayesian inversion[44–46].

According to Bayes' theorem

$$P(\mathbf{m}|\mathbf{d}) = \frac{P(\mathbf{m})\, P(\mathbf{d}|\mathbf{m})}{P(\mathbf{d})} = \frac{P(\mathbf{m})\, \mathcal{L}(\mathbf{m})}{\int_{\mathcal{M}} P(\mathbf{m})\, \mathcal{L}(\mathbf{m})\, d\mathbf{m}}, \tag{3}$$

where $P(\mathbf{m}|\mathbf{d})$ is the PPD of a vector $\mathbf{m}$ of $M$ model parameters given a vector $\mathbf{d}$ of $N$ data, $P(\mathbf{m})$ is the prior density representing probabilistic a priori model parameter information; $P(\mathbf{d}|\mathbf{m})$ is the conditional probability of $\mathbf{d}$ given $\mathbf{m}$, which can be interpreted as the likelihood of $\mathbf{m}$ given $\mathbf{d}$ once $\mathbf{d}$ is observed, written as $\mathcal{L}(\mathbf{m})$; and $P(\mathbf{d})$ is a normalization term known as Bayesian evidence, which is an integral over the state space $\mathcal{M}$.

The trans-D approach includes the number of parameters as an unknown, in this case the number of sediment layers $k$. The resulting posterior spans multiple spaces of different dimensions and includes the uncertainty due to limited knowledge of the model parameterization. Formulating equation (3) as a Bayesian hierarchical model including $k$ as shown by[44], the trans-D joint posterior is

$$P(k, \mathbf{m}_k|\mathbf{d}) = \frac{P(k)P(\mathbf{m}_k|k)P(\mathbf{d}|k, \mathbf{m}_k)}{\sum_{k' \in \mathcal{K}} \int_{\mathcal{M}} P(k')P(\mathbf{d}|k', \mathbf{m}'_{k'})P(\mathbf{m}'_{k'}|k')d\mathbf{m}'_{k'}}, \quad (4)$$

where $k \in \mathcal{K}$ ($\mathcal{K}$ being a countable set) indexes possible model choices and $P(k)$ is the prior over the $\mathcal{K}$ models considered.

Fixed-dimensional PPDs are typically sampled using the Markov chain Monte Carlo algorithm (MCMC) which is a numerical approximation technique to conveniently sample from complex and high-dimensional functions in Bayesian statistics[47]. Given a current set of model parameters $\mathbf{m}$, a new set $\mathbf{m}'$ is drawn from a proposal density $Q(\mathbf{m}'|\mathbf{m})$ and either accepted to the Markov chain or rejected based on comparing an acceptance probability $A(\mathbf{m}'|\mathbf{m})$ to a uniform random number on [0,1]. If $\mathbf{m}'$ is rejected, another copy of $\mathbf{m}$ is added to the Markov chain.

Extending this for trans-D parameter spaces, the reversible jump MCMC (rjMCMC) with the Metropolis-Hastings-Green acceptance criterion is used here[44]. The acceptance probability is

$$A(k', \mathbf{m}'_{k'}|k, \mathbf{m}_k) = \min\left[1, \frac{Q(k, \mathbf{m}_k|k', \mathbf{m}'_{k'})}{Q(k', \mathbf{m}'_{k'}|k, \mathbf{m}_k)} \frac{P(k')}{P(k)} \frac{P(\mathbf{m}'_{k'}|k')}{P(\mathbf{m}_k|k)} \frac{\mathcal{L}(k', \mathbf{m}'_{k'})}{\mathcal{L}(k, \mathbf{m}_k)} |\mathbf{J}|\right], \quad (5)$$

where $P(k)\,P(\mathbf{m}_k|k)$ and $\mathcal{L}(k, \mathbf{m}_k)$ represent the prior and likelihood, respectively, for model choice $k$ and corresponding parameters $\mathbf{m}_k$, and $|\mathbf{J}|$ is the determinant of the Jacobian matrix for the transformation from state $(k, \mathbf{m}_k)$ to $(k', \mathbf{m}'_{k'})$, which is equal to one in this implementation.

As is common in geophysical inverse problems, a nonlinear system response and a potentially multi-modal solution space can strongly reduce the efficiency of the basic MCMC method. To improve sampling posterior modes and achieve efficient proposals for the dimension-jump steps of the rjMCMC algorithm, we employ the population sampling approach known as parallel tempering by drawing samples from additional intermediate distributions using parallel interacting chains with successively relaxed likelihoods[48]. This utilizes the parallel computing capabilities of clustered CPUs and allows fast and efficient convergence within the multi-dimensional nonlinear space. In our implementation, each CPU manages one Markov chain, while the GPUs carry out the forward model calculations.

Applying single-parameter perturbations to the parameters in a Markov-chain move using the common choice of multi-variate Gaussian proposal densities can become inefficient at high dimensions. It has been shown in fixed-D and trans-D inversions that applying perturbations in a principal-component (PC) space, where PC parameters are uncorrelated, can improve sampling efficiency by an order of magnitude and more for some problems. We apply perturbations to PC transformed parameters based on decomposition of the unit-lag covariance matrix with a different PC decomposition applied for each chain and number of interfaces $k$ as described in[49].

## Likelihood function

To evaluate equation (5) in a sampling algorithm, the likelihood function $\mathcal{L}$ must be specified. The likelihood is based on the residual error distribution described by $P(\mathbf{d}|\mathbf{m})$ for residuals $\mathbf{r}(\mathbf{m}) = \mathbf{d} - \mathbf{d}(\mathbf{m})$ of observed data $\mathbf{d}$ and

predicted data $\mathbf{d}(\mathbf{m})$. Residuals are commonly assumed to be Gaussian distributed due to the central limit theorem, and $P(\mathbf{d}|\mathbf{m})$ is then formulated as a multivariate Gaussian function. The conditional probability $P(\mathbf{d}|\mathbf{m})$ is interpreted as the likelihood function $\mathcal{L}(\mathbf{m})$ of $\mathbf{m}$ for fixed $\mathbf{d}$, and is given by

$$\mathcal{L}(\mathbf{m}) = \frac{1}{(2\pi)^{N/2} \prod_{i=1}^{N_D} \prod_{j=1}^{N_i} \sigma_{ij}} \exp\left[-\frac{1}{2} \sum_{i=1}^{N_D} \sum_{j=1}^{N_i} \left(\frac{r_{ij}(\mathbf{m})}{\sigma_{ij}}\right)^2\right], \quad (6)$$

where $N$ is the number of data consisting of $N_D$ frequency-grouped subsets indexed by $i$, with data $d_{ij}$ and standard deviations $\sigma_{ij}$ for $j = 1, \ldots, N_i$ angles in the $i$th subset. Generally, a covariance matrix $\mathbf{C}$ is required in the multivariate Gaussian density function, but in this work, we assume uncorrelated errors such that $\mathbf{C}$ is a diagonal matrix with variances on the main diagonal. We consider the data spacings in frequency and angle to be sufficient that error correlations are negligible, and acknowledge the large challenges in estimating error covariances in strongly nonlinear inversion — particularly when such a large number of inversions ( ~ 1700) are carried out. For these reasons, we fixed all $\sigma_{ij}$ to match the standard error about the mean of 11 neighboring CDPs for each angle and frequency with a multiplicative factor determined by trial and error: 2.5 for the lower 4 frequency bands and 1.5 for the upper 4 bands. These error assumptions resulted in test inversions that produced geoacoustic models with well-constrained structure and about 8-10 layers on average, which we consider reasonable for this sedimentary environment. Without the error adjustment, inversions gave poorly-constrained models with more layers (>14 on average), including multiple thin layers with parameter values that differed significantly from surrounding layers – this appears to be spurious structure introduced by over-fitting the data (i.e., setting standard deviations that are too small). Increasing the fixed data variance values is a conservative choice to reduce the chance of over-fitting the data, while inversions for $\mathbf{C}$ can be unstable due to the highly non-linear model becoming too under-determined. It is thus a pragmatic solution to an otherwise currently intractable problem.

## Prior distributions

Prior information constrains the posterior solution, which is important for nonlinear problems that can err into unrealistic solution spaces. Both independent and conditional prior distributions are used, with their parameter values listed in the tables of the supplementary document. The independent prior distributions are uniform over a chosen interval (Supplementary Table 1). The conditional priors are uniform over a defined area and are used to constrain parameters that often correlate in nature—porosity $\beta$ and compressional grain-grain modulus $\gamma_P$ as $P(\beta|\gamma_P)$ (Supplementary Table 2) and compressional sound speed $c_p$ and density $\rho$ as $P(c_p|\rho)$ (Supplementary Table 3). The conditional prior distribution of $c_p$ and $\rho$ is based on laboratory and in situ measurements[38,50] and has been used in previous studies with slight modifications[51,52]. It consists of the area between empirical lower and upper bound curves for $c_p$ as a function of $\rho$, with each curve of the form

$$c_p = 1000(g_1 - g_2\rho + g_3\rho^{g_4})g_5, \quad (7)$$

where $g_1$ to $g_5$ are fit parameters. In addition, the conditional prior distribution of $\beta$ and $\gamma_P$ has been chosen to suggest at least a weak correlation of $c_p$ and $\rho$ while avoiding highly unlikely parameter value combinations.

## Model assumptions

In summary, our approach makes the following assumptions. The ASR correction for the bottom-reflected path uses the specular angle although sub-bottom returns contribute reflected energy at steeper angles. The 2D track is considered to consist of local 1D models. Each 1D model is constructed by horizontal, homogeneous layered media with a variable number of layers above a lower half space. Scattering from interfaces or inhomogeneities are not modeled. The GS model is assumed to represent

**Article**

unconsolidated marine sediments while ignoring classical viscosity effects. We do not infer shear wave parameters and assume compression to shear conversion to be small. The bulk moduli and densities of grains and pore fluid are fixed. The RC data are modeled as independent variables with fixed, normally distributed standard deviations based on population subset statistics. The parameters' PPDs are modeled with bounded, uniform prior distributions and model residuals are assumed to be normally distributed.

## Data availability

The reflection coefficient data are available at the following Figshare.com repository: https://doi.org/10.6084/m9.figshare.23820105.

## Code availability

The codes for carrying out Bayesian inversion are avialable upon request to Jan Dettmer (jan.dettmer@ucalgary.ca). Figures were produced with Matplotlib[53], the Generic Mapping Tools[54], and MATLAB.

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

## Acknowledgements

We gratefully acknowledge the support of an Office of Naval Research grant (Grant No. N00014-20-S-B001). The measurements were made possible by the CLUTTER JRP, a collaboration among ARL-PSU (U.S.), DRDC-A (Canada), NATO Undersea Research Center (Italy), and NRL (U.S.). The computational work was carried out on parallel high-performance computing clusters, one operated by the authors at the University of Victoria funded by the Natural Sciences and Engineering Research Council of Canada and the Office of Naval Research, and three clusters of the Digital Research Alliance of Canada (alliancecan.ca): Cedar (Simon Fraser University, BC DRI Group), Graham (University of Waterloo, Compute Ontario) and Beluga (McGill University, Calcul Québec).

## Author contributions

C.W.H. conceived the project, designed the experiment, collected the data, processed them for the frequency-domain reflection coefficient and wrote the forward model codes. T.S. carried out the spectral data processing, inversion, and data analyses. T.S. wrote the manuscript, while J.D., C.W.H and S.E.D. contributed subsections. All authors contributed to the editing and refinement of the final manuscript.

## Competing interests

The authors declare no competing interests.

## Additional information

**Peer review information** : *Communications Engineering* thanks Tsu Wei Tan, David R. Dall'Osto and the other, anonymous reviewer for their contribution to the peer review of this work. Primary Handling Editors: Anastasiia Vasylchenkova, Rosamund Daw. A peer review file is available.

