## [Peer Review File · Communications Engineering]

Reviewers' comments:

Reviewer #1 (Remarks to the Author):

The authors present a geoacoustic inversion method applied to data collected in 2009 by a single autonomous underwater vehicle equipped with a source ranging from 800 to 3500 Hz and a 32-hydrophone array. These non-real-time inferred seabed properties have a resolution of approximately 0.1 m in the vertical and 50 m in the horizontal. The results of this mesoscale, subsurface seabed quantification up to approximately 6 m largely agree with the seismic core samples at the experimental site off the coast of Sicily, as well as with previous work using the same dataset in Ref. [25].

Although the data itself is not new, the methods combining reflection coefficient and trans-D seabed quantification techniques are original. The inversion results are highly convincing. The methodology applied in this work and the overall workflow are organized, well-presented, and properly referenced. With the data and code available after publication, this methodology is poised to have a substantial impact on the underwater acoustic sensing community. My comments for minor revisions are as follows.

Conclusion (P. 13)

Line 26-27: "It is also expected to advance understanding of sound propagation, hence ambient noise..." The discussion of the scattering effects of AUV and their impact on inversion seems more appropriate for future research. Other sources of ambient noise, such as wind, rain, marine life, and human activities like ship traffic, are not addressed in this work. Therefore, the phrase "understanding ...ambient noise" should be excluded from the Conclusion section.

Methods (P. 14-16)

The data used in this study and the seabed reflection coefficient (RC) method are based on the authors' previously published article in Ref[25], with an error correction applied in the reflection data processing and consideration of the effect of AUV scattered return (ASR). The inversion is carried out using a robust trans-D method as detailed in Refs [35, 36, 43-45, 48, 50, 51], etc.

Without normalizing the bottom reflected path P_r (page 15, line 20), it is unclear how ASR affects the trans-D inversion results presented in Figs 3, 4, and 5(a). Additionally, it is not clear what significant improvements are achieved by the updated Eq. (1) for the different pings 2753, 2755, and 2757, as shown in Figs 10 and 11 in Ref [25]. Would this normalization support the statement in **Likelihood function** section that the covariance matrix \mathbf{C} is diagonal based on the assumption that errors are uncorrelated (page 20, line 3-6) ? It would be more persuasive to provide information on how ASR removal improves the inversion results by mentioning the results without normalizing the bottom reflected path P_r .

Methods: Modeling reflection coefficient data (P.17)

Line 21: "homogeneous layers above a lower half space as shown in Fig. 1" - Should this reference be to Figure 2a rather than Fig 1?

Methods: Bayesian Interference, Likelihood function and Prior Distribution (P.17-20)

Prior information plays a crucial role in determining a feasible posterior solution. It would be helpful to clarify whether the chosen upper and lower bounds in **Supplementary Tables 1–3** cover the geoacoustic properties previously inferred at Site 2, as shown in Table II of Ref.[25], for both independent and conditional prior distributions.

Minors:

Page 7, line 30: "while our our" ... There is an extra "our"

Key Points: "We present an automatic geoacoustic inversion method" The data processing section reveals a complicated regularization is necessary, maybe "automatic" is not proper?

Reviewer #2 (Remarks to the Author):

This paper extends prior work and uses modern computing resources to invert a well published dataset (e.g Ref 25). It demonstrates how to invert this data for sediment properties like porosity without biasing towards a model with fixed layers or apriori information on stratigraphy. The statistical analysis is rigorous.

One question regarding the statement in the introduction (Line 18), "...high-resolution chirp sonar, marine seismic reflection, and refraction surveys¹¹. None of these approaches provide both sufficient resolution and material properties required for next-generation survey applications"

Why can't this same process be applied to seismic reflection/refraction surveys? It seems any broadband repetitive signals received on a trailing array would be amenable to this approach and the statement seems to make it apply only to AUV data.

Additional comments/notes are in annotated PDF attachment

Reviewer #3 (Remarks to the Author):

It is a manuscript with good scientific quality. It presents original geoacoustic inversion where the trans-dimensional Bayesian inference is carried out over data obtained by autonomous underwater vehicle for. And the scale of about 10 cm and 50 m in the vertical and horizontal is achieved along a 12 km survey track. Specially, porosity as one of the inferred physical parameters is discussed in detail, which is seldomly involved in traditional geoacoustic inversion. The inverted parameters are validated by coring measurements.

There are some questions needed to be further explained before it is accepted for publication.

1. The noise emitted by AUV may be an inference in the received data which is overlapped with the seabed reflection. How to deal with the emission noise in geoacoustic inversion?
2. Line27, Eq.2, "where θ the seabed grazing angle"->"where θ is the seabed grazing angle".

Response to Reviewers

We thank all reviewers for their comments, and we have improved the manuscript accordingly. Please note that we have updated the draft line number references as well for clarity.

Reviewer 1

Conclusion (P. 13)

“Line 26-27: "It is also expected to advance understanding of sound propagation, hence ambient noise..." The discussion of the scattering effects of AUV and their impact on inversion seems more appropriate for future research. Other sources of ambient noise, such as wind, rain, marine life, and human activities like ship traffic, are not addressed in this work. Therefore, the phrase "understanding ...ambient noise" should be excluded from the Conclusion section.”

Response:

We agree and removed “hence ambient noise”. (Lines 269-270)

Methods (P. 14-16)

“The data used in this study and the seabed reflection coefficient (RC) method are based on the authors' previously published article in Ref[25], with an error correction applied in the reflection data processing and consideration of the effect of AUV scattered return (ASR). The inversion is carried out using a robust trans-D method as detailed in Refs [35, 36, 43-45, 48, 50, 51], etc.

Without normalizing the bottom reflected path P_r (page 15, line 20), it is unclear how ASR affects the trans-D inversion results presented in Figs 3, 4, and 5(a). Additionally, it is not clear what significant improvements are achieved by the updated Eq. (1) for the different pings 2753, 2755, and 2757, as shown in Figs 10 and 11 in Ref [25]. Would this normalization support the statement in Likelihood function section that the covariance matrix C is diagonal based on the assumption that errors are uncorrelated (page 20, line 3-6) ? It would be more persuasive to provide information on how ASR removal improves the inversion results by mentioning the results without normalizing the bottom reflected path P_r .”

Response:

The updated Eq.(1) reduces some artifacts in the data and therefore has some influence on the inversion results. However, it is not easy to isolate its effect on the inversion results compared to the numerous other factors that play into convergence and stability on an individual ping or groups of pings. Thus, we can confidently say that the data are improved by Eq(1), but quantifying the significance in the inversions is not easy. Another way to say this is that we think it is an important correction to the method by reducing data artifacts, but it may not have a significant impact on this data set.

In order to address the reviewer’s concern, on Line 323, we have replaced the sentence “This greatly reduced the artifacts.”

to

“This clearly reduced the artifacts in the data, which is important for the method generally. However, the specific impact of this processing on inversion results is not quantified in this study.”

Methods: Modeling reflection coefficient data (P.17)

“Line 21: "homogeneous layers above a lower half space as shown in Fig. 1" - Should this reference be to Figure 2a rather than Fig 1?”

Response:

Thank you, we have corrected the reference to Figure 2a. (Line 392)

Methods: Bayesian Interference, Likelihood function and Prior Distribution (P.17-20)

“Prior information plays a crucial role in determining a feasible posterior solution. It would be helpful to clarify whether the chosen upper and lower bounds in Supplementary Tables 1–3 cover the geoacoustic properties previously inferred at Site 2, as shown in Table II of Ref.[25], for both independent and conditional prior distributions.”

Response:

The values for Site 2 in Table II of Ref.[25] were hypothetical assumptions for an ad hoc forward simulation in that previous study. However, it is important to clarify what the chosen prior boundaries contain, and we have added the following sentence to the prior section of the supplemental document: “These prior bounds encompass all marine unconsolidated sediments.” (Line 55 Suppl.)

Minors:

“Page 7, line 30: “while our our” ... There is an extra “our””

Response: Fixed (Line 170)

“Key Points: “We present an automatic geoacoustic inversion method” The data processing section reveals a complicated regularization is necessary, maybe “automatic” is not proper?”

Response: Changed to “semi-automatic”. (Line 9)

Reviewer 2

“One question regarding the statement in the introduction (Line 18), "...high-resolution chirp sonar, marine seismic reflection, and refraction surveys¹¹. None of these approaches provide both sufficient resolution and material properties required for next-generation survey applications”

Why can't this same process be applied to seismic reflection/refraction surveys? It seems any broadband repetitive signals received on a trailing array would be amenable to this approach and the statement seems to make it apply only to AUV data.”

Response:

The reviewer makes a good point. This method does not require an AUV. Data from a suitable (but not any) broadband repetitive source received on a suitable (but not any) trailing array with a suitable (but not any) geometry design can be used for the inversion method described here.

Rather than to attempt to describe all the limitations of typical marine seismic survey designs for this method, and to properly broaden the scope of the data collection as the reviewer rightly points out, we make the following changes:

Line 37: Changed ‘None of these approaches provide...’ to ‘None of these approaches **have provided to date...**’

Lines 63-70: Changes (in red):

We present a method for meso-scale geoacoustic seabed quantification (MGSQ) through automated analysis of survey data **from AUVs** that adapts a seabed model to the structure resolved by the data. The method greatly simplifies seabed surveys by reducing time, cost, and subjective operator choices in producing seabed images. We collected acoustic data along a 12 km track on the Malta Plateau in the Mediterranean Sea (Fig. 1) using an AUV towing an impulsive acoustic source and a linear array of 32 hydrophones (Fig. 2b). **Even though an AUV was employed here, the method is equally applicable to towed sources and arrays that** can record direct and bottom-reflected arrivals for a large number of source transmissions (pings) along the survey track. **Care must be taken with respect to position of source and array in the water column to ensure that direct and bottom-reflected paths can be separated.**

“p.7,L29: Is there a reference that could inform the reader on this device (boomer source)?”

Response:

We have added a reference [27] which contains a good description of that device. (Line 169)

“p.17,L7: suggest summarizing constraints in discussion section - data processing - GS model - frequencies in inversion - angular resolution missing critical angle”

Response:

We have added a paragraph about the most important model assumptions in the method section at the end (Lines 492-502):

“In summary, our approach makes the following assumptions. The ASR correction for the bottom-reflected path uses the specular angle although sub-bottom returns contribute reflected energy at steeper angles. The 2D track is considered to consist of local 1D models. Each 1D model is constructed by horizontal, homogeneous layered media with a variable number of layers above a lower half space. Scattering from interfaces or inhomogeneities are not modeled. The GS model is assumed to represent unconsolidated marine sediments while ignoring classical viscosity effects. We do not infer shear wave parameters and assume compression to shear conversion to be small. The bulk moduli and densities of grains and pore fluid are fixed. The RC data are modeled as independent variables with fixed, normally distributed standard deviations based on population subset statistics. The parameters’ PPDs are modeled with bounded, uniform prior distributions and model residuals are assumed to be normally distributed.”

“P.17,L23: After sentence: "Due to the proximity of source and receiver to the seabed in the survey considered here, the plane-wave assumption is invalid and would lead to significant errors in modeling the observed RC values.", commented: "near the critical angle".”

Response:

Certainly at low frequencies spherical wave effects are important near the critical angle. However, spherical wave effects can be important at other steeper angles. Consider for simplicity, the case of a source and receiver Y m above a layer Y m thick. At grazing angles less than normal incidence, the spherical wave reflection coefficient could be roughly approximated with two eigenrays, one striking the water-sediment interface at the specular angle and one striking the lower layer at some other (steeper) source angle. Even far above the critical angle, those two angles can be quite different, leading to substantive differences with plane-wave reflection for which both eigenray source angles are by definition identical. In summary, spherical wave effects are important a) near the critical angle AND b) when the total layer thickness is comparable to the source/receiver height above the bottom. In summary, our statement is correct as it stands. We don't need to invoke a critical angle.

“P.11, L23: Can you place an approximate depth of this unit 1 and unit 2?”

Response:

Due to the time windowing chosen, the data do not represent much of unit 2. However, we have added this sentence regarding the previously inferred thickness of unit 1 by ref.[33]:

“Unit 1 is bounded by an erosional horizon previously detected with a thickness of 5 to 10 ms two way travel time (3.75 to 7.5 m at 1500 m/s).” (Lines 206-208)

“P.19,L20: Does each model have its own principal component space?”

Response:

Yes, each Markov chain and each model parametrization, i.e. numbers of interfaces k , gets its own PC decomposition based on Jacobian matrices of Markov chain samples. They are also updated during the burn-in stage. We added the following to that sentence:

“... with a different PC decomposition applied for each chain and number of interfaces k ...” (Lines 452-453)

Typos and minor edits

“well-being” (Line 29)

“detection” (Line 33)

removed “of” (Line 251)

Reviewer 3

“1. The noise emitted by AUV may be an interference in the received data which is overlapped with the seabed reflection. How to deal with the emission noise in geoaoustic inversion?”

Response:

The noise emitted by the AUV (as well as other noise sources) is mitigated by the use of linear frequency modulated (LFM) pulse. By match-filtering the pulse, all other noise is reduced that has different spectral or temporal characteristics than the LFM.

We added the abbreviation in line 283 and one sentence in lines 284-285:

“The LFM pulses have good noise rejection characteristics; by match-filtering the data, all noise is reduced that exhibits different spectral or temporal characteristics than the LFM. ”

“2. Line27, Eq.2, “where Θ the seabed grazing angle”->“where Θ is the seabed grazing angle”.”

Response: Fixed. (Line 397)

Finally, we found a few sentences that needed additional clarification.

In lines 372 – 380, we have replaced

" To resolve τ , clear observation of dispersion in both sound speed and attenuation over a sufficient frequency and angular range is required. The AUV data do resolve sound speed, but the small Bragg pattern deviations indicating dispersion are not sufficiently evident (with the limited frequency and angular range) to constrain τ . “

With

" To resolve τ , a clear observation of classical viscosity in the sound speed dispersion and/or frequency-dependent attenuation is required. For example, classical viscosity leads to an attenuation that increases with frequency, f , as f^2 squared at low frequencies and $f^{\frac{1}{2}}$ at high frequencies. However, the data do not show such behavior. This could be either because classical viscosity plays a negligible role in these sediments over this frequency range, or there is an insufficient frequency and angular range to observe it. For either case, the inversion is best served by using the GS model. “

Reviewers' comments:

Reviewer #1 (Remarks to the Author):

The authors present a geoacoustic inversion method applied to data collected in 2009 using a singular autonomous underwater vehicle equipped with a source spanning from 800 to 3500 Hz and a 32-hydrophone array. The post-processed inferred seabed properties exhibit a resolution of approximately 0.1 m vertically and 50 m horizontally. The outcomes of this mesoscale, subsurface seabed quantification, reaching depths of approximately 6 m, align notably with seismic core samples taken at the experimental site off the coast of Sicily. Furthermore, these results correlate well with prior studies employing the same dataset (Ref. [25]).

While the data itself is not novel, the innovation lies in the novel methods combining reflection coefficients and trans-D seabed quantification techniques, showcasing the considerable potential for real-time seabed probing. The inversion results are highly compelling. The methodology employed in this study and the overall workflow are very organized, well-presented, and appropriately referenced, and questions are answered after reviewing.

Moreover, the availability of both data and code post-publication enhances the impact of this methodology within the underwater acoustic sensing community.

Reviewer #2 (Remarks to the Author):

The paper could use a bit more discussion on the limitations of the method upfront, as the heavy graphics need to be interpreted with caution. The conclusion holds that this will be useful for archeology or explosive ordinance remediation, but really provides a better description of sediment types and layers but is not sensitive to occlusions (those suggested by the data are likely False Alarms from the 1D approximation as discussed in section starting line 234)

Below are a few points/suggestions that may make the paper stronger. Starting with the introduction, it would be nice to establish a flow to the paper. AUVs can measure sub-bottom properties in upper sediments. An example is given to discuss the data, and acknowledge limitations of that dataset (e.g. no low-angle results) can be overcome with better experimental design.

The second paragraph (lines 41-47) neatly introduces remote sensing, and should be the leading paragraph.

It would transition nicely with an additional final sentence (for example, "AUVs equipped with sonars and hydrophone arrays are ideal platforms for sub-bottom characterization and autonomous survey. [ref. 25]")

Comment on Fig. 2A. The ray paths show a downward refraction, implying all the layers are slow sediment layers.

It may make sense to call these "rays" out specifically in the text around Line 80, as to not confuse them with energy pathlines.

Line 118: Define "This layer" is it the top most layer?

Suggest reordering: "The top mud layer (0.8 porosity) decreases in thickness from north to south (left to right in Fig. 3) with an initial thickness of 1.2 m that reduces to less than 0.2 m and possibly even disappears."

to come after the following sentence "referred to as the mud wedge"

Line 127-133: Start a new paragraph. The reader wants to know why the high porosity in the basement (if confidence low, should that result be windowed out?)

Line 144: Restate the top two layers are the "mud wedge"

Line 146: Is attenuation driven by the grain-to-grain modulus? Perhaps move this sentence after the next statement which introduces the difference. "This trend is expected inasmuch..."

Line 150: Unclear if the material index n the "sediment -to-water sound speed ratio defined on line 142?) Please define " n " or if not needed omit the variable.

Line 187: Where should the reader "note the small uncertainties" ? Is that what the following sentence explains at 65 degrees?

Line 202: Maybe state these were 'deeper cores' as to sample this erosional layer.

Line 213: Perhaps soften statement, "The present study suggests isolated pockets of dense material in the swales of the erosional layer, interpreted here as coarse lag sediments that would tend to be deposited there. Alternatively, these detections may be artifacts due to acoustic focussing effects (see section on Limitations regarding local 1D model assumption)."

Line 218: Define some of the "seabed characteristics" in this sentence. (e.g. sediment volume heterogeneities as discuss on 225, or layering causing resonant conditions on line 229)

Line 347: This statement is buried in the methods, but should be held more prominently in the discussion.

Line 488: If I interpret this correctly, the conditional prior is that for a given sound speed the density is bounded by the curves. This limits the search space of parameters. Does the inversion produce results that are within the limits of these prior distributions, or at the extrema? Is the performance negatively affected by reducing the bounds?

Reviewer #3 (Remarks to the Author):

The revised manuscript needs to be modified before accept for publication. The following questions or problems are needed to be further answered or solved in the new manuscript.

1. Line9, " We present a semi-automatic geoacoustic inversion". However , Line18, "We present an automated geoacoustic inversion method ".
2. Line60, " horizontal variability at $O(10^0)$ to $O(10^3)$ m, which we term geoacoustic meso-scale variability". In fact, $O(10^0)$ to $O(10^3)$ m is referred to as small size.
- 3.Line69-70, "Care must be taken with respect to position of source and array in the water column to ensure that direct and bottom-reflected paths can be separated".How to realize it?
4. Line79, "bulk density, sound speed, and sound attenuation (the latter two are frequency dependent)". As we know, sound speed is independent of frequency.
5. Line78-79, "The GS parameters can be transformed to sediment parameters of bulk density, sound speed, and sound attenuation" should be provided by some references to further explain it.
- 6.Line127, "Structures deeper than this denser layer have higher uncertainties" is difficult to be understood.
7. Line285, "The LFM pulses have good noise rejection characteristics; by match-filtering the data, all noise is reduced that exhibits different spectral or temporal characteristics than the LFM". It is difficult to understand the sentence.

Response to Reviewers

We thank all reviewers for their comments, and we have improved the manuscript accordingly. The line numbers of the updated manuscript PDF and its annotated PDF have slightly shifted compared to the previous version due to minor changes.

Manuscript edits are indicated in the file “manuscript_diff_marked2.pdf”. As before, removed text is given in red and strike-through markings, while new text is shown in blue. The page rims contain gray bars for any changed lines.

In this response, original reviewer text is set in black, while our response is set in blue. The updated line numbers and reference numbers (one new ref added) are also given if they are different (in red parentheses), in addition to the old original numbers. The updated numbers refer to the PDF with difference markups.

Reviewer 1

The authors present a geoacoustic inversion method applied to data collected in 2009 using a singular autonomous underwater vehicle equipped with a source spanning from 800 to 3500 Hz and a 32-hydrophone array. The post-processed inferred seabed properties exhibit a resolution of approximately 0.1 m vertically and 50 m horizontally. The outcomes of this mesoscale, subsurface seabed quantification, reaching depths of approximately 6 m, align notably with seismic core samples taken at the experimental site off the coast of Sicily. Furthermore, these results correlate well with prior studies employing the same dataset (Ref. [25]).

While the data itself is not novel, the innovation lies in the novel methods combining reflection coefficients and trans-D seabed quantification techniques, showcasing the considerable potential for real-time seabed probing. The inversion results are highly compelling. The methodology employed in this study and the overall workflow are very organized, well-presented, and appropriately referenced, and questions are answered after reviewing.

Moreover, the availability of both data and code post-publication enhances the impact of this methodology within the underwater acoustic sensing community.

Thank you, we have addressed all points of the first revision.

Reviewer 2

The paper could use a bit more discussion on the limitations of the method upfront, as the heavy graphics need to be interpreted with caution. The conclusion holds that this will be useful for archeology or explosive ordinance remediation, but really provides a better description of sediment types and layers but is not sensitive to occlusions (those suggested by the data are likely False Alarms from the 1D approximation as discussed in section starting line 234)

We have removed the sentence about archeology and unexploded ordnance to improve clarity. As to the limitations, see our comment further below about “Line 347”.

Below are a few points/suggestions that may make the paper stronger. Starting with the introduction, it would be nice to establish a flow to the paper. AUVs can measure sub-bottom properties in upper sediments. An example is given to discuss the data, and acknowledge limitations of that dataset (e.g. no low-angle results) can be overcome with better experimental design.

As to the comment on no low angles, we address this below, “Line 347”.

The second paragraph (lines 41-47) neatly introduces remote sensing, and should be the leading paragraph.

It would transition nicely with an additional final sentence (for example, "AUVs equipped with sonars and hydrophone arrays are ideal platforms for sub-bottom characterization and autonomous survey. [ref. 25]"

We choose to keep the current sentence ordering in accordance with previous reviews.

Comment on Fig. 2A. The ray paths show a downward refraction, implying all the layers are slow sediment layers.

It may make sense to call these "rays" out specifically in the text around Line 80, as to not confuse them with energy pathlines.

To improve clarity, we have edited two labels in Fig. 2A to “seabed reflected ray” and “subbottom reflected ray”, and added “... illustrated by ray paths.” to the figure caption.

Line 118: Define "This layer" is it the top most layer? (Line 122)

Clarified to “This second layer ...”, as it continued describing the next layer from top down.

Suggest reordering: "The top mud layer (0.8 porosity) decreases in thickness from north to south (left to right in Fig. 3) with an initial thickness of 1.2 m that reduces to less than 0.2 m and possibly even disappears." to come after the following sentence "referred to as the mud wedge"

Reordering is not necessary here, due to an avoided misunderstanding by the previous clarification.

Line 127-133: Start a new paragraph. The reader wants to know why the high porosity in the basement (if confidence low, should that result be windowed out?)

New paragraph has been started now (L. 131-132). However, it is an intentional decision not to window out results, as we present the full uncertainty quantification in Supplemental Fig. 2 in order to report results objectively.

Line 144: Restate the top two layers are the "mud wedge"

Thank you, we have restated that the upper two layers are the mud wedge (L. 149), and removed a newly redundant statement two sentences further down (L. 152).

Line 146: Is attenuation driven by the grain-to-grain modulus? Perhaps move this sentence after the next statement which introduces the difference. "This trend is expected inasmuch..."

The relationship between compressional grain-to-grain modulus and attenuation is sufficiently complex that describing it here would deviate too much from the point of reporting the observed trend. A paper concerning the practical meaning of the GS parameters in relation to measured data is given as reference 26. We have added the reference to that sentence now (L. 154). However, a full description of the physical and mathematical relationship of all parameters in the viscous grain shearing model is given in references 38 and 39 (upd.: 39 and 40).

Line 150: Unclear if the material index n the "sediment -to-water sound speed ratio defined on line 142?) Please define " n " or if not needed omit the variable. (L. 154)

" n " is the material index. Now, we have replaced its symbol with "material index" everywhere in this manuscript to improve clarity. This parameter affects the frequency-dependent behavior of attenuation and sound speed. The values and uncertainty of " n " are reported in the Supplemental material. Edited lines: (L. 154, 378, 393)

Line 187: Where should the reader "note the small uncertainties" ? Is that what the following sentence explains at 65 degrees?

Relatively small uncertainties are observed for the interface depth values, which are indicated in the leftmost panel of Fig. 4 a and b (for two RC data sets), and Supplement Fig. 2, where the parameter values' uncertainty is only slightly elevated narrowly along interfaces. We have added "... as indicated in Fig. 4 a and b ..." to the text. (L. 192, 192)

Line 202: Maybe state these were 'deeper cores' as to sample this erosional layer.

Thank you, we changed it to "... deeper cores ..." (L. 208)

Line 213: Perhaps soften statement, "The present study suggests isolated pockets of dense material in the swales of the erosional layer, interpreted here as coarse lag sediments that would tend to be deposited there. Alternatively, these detections may be artifacts due to acoustic focussing effects (see section on Limitations regarding local 1D model assumption)."

The manuscript does not state "interpreted here", instead, it gives a softer statement of "These could be detections of ...". However, we have changed "detections" to "occurrences" to further indicate that we do not claim explicit detections. (L. 220)

Line 218: Define some of the "seabed characteristics" in this sentence. (e.g. sediment volume heterogeneities as discuss on 225, or layering causing resonant conditions on line 229)

We have changed "Seabed characteristics" to "Geoacoustic properties" (L. 225)

Line 347: This statement is buried in the methods, but should be held more prominently in the discussion.

Three comments (the reviewer's paragraphs 1 and 2 "discussion of limitations", and on "Line 347") indicate the reviewer may believe the lack of low grazing angle data presents a serious problem for the data set and manuscript.

We do not believe this to be the case. The measured data we present covers an angular range from 32 – 67 degrees. Over this angular range, combined with the frequency range we employ, the information content about seabed properties is extremely high, as manifest from results in Figures 3-5. There is no need for ‘caution’ to the reader, on the contrary, the spatial resolution of key sediment properties obtained along with the uncertainties clearly indicate significant geoaoustic information in the data.

Furthermore, we already specifically address the limited angular aperture in Lines 179-183 (L. 185-189), noting that the geoaoustic information content is jointly shared between the angular and the frequency domains (as dictated by the Bragg condition) and that a broad frequency range can partially compensate for the modest angle range.

The point of how the frequency domain can compensate for the angular domain is further discussed in the Methods section, Lines 347 – 350 (L. 349-358) in a simulation study. Those simulation results belong in the Methods section. They are not 'buried'.

For clarity, we have added two additional sentences:

On line 70 (L. 71), concerning the data content and its relationship to physical properties:

“...(CDP) gathers. The specific experiment we employ permits processing of seabed reflection-coefficient data between 32 and 67 degrees and between 900 and 3400 Hz. We will demonstrate that these data can provide detailed knowledge about geoaoustic properties which is contained in the Bragg interference pattern of the reflection coefficient as a function of angle and frequency. The data...”

And following line 183 (in L. 190):

“An example of this is given in the Methods section.”

Line 488: If I interpret this correctly, the conditional prior is that for a given sound speed the density is bounded by the curves. This limits the search space of parameters. Does the inversion produce results that are within the limits of these prior distributions, or at the extrema? Is the performance negatively affected by reducing the bounds?

Improving clarity, we now specifically mention in Line 478 (L. 486-487) that the two conditional priors “are uniform over a defined area” as given in Supplementary Tables 2 and 3.

These prior distributions were chosen to restrict the search space to physically plausible values of known sediments based on published datasets. Further reducing the bounds would eliminate desirable plausible solutions and is not recommended.

Reviewer 3

The revised manuscript needs to be modified before accept for publishment. The following questions or problems are needed to be further answered or solved in the new manuscript.

1. Line9, “ We present a semi-automatic geoaoustic inversion”. However , Line18, “We present an automated geoaoustic inversion method ”.

Thank you, we adjusted that “automated” to “semi-automated” in that sentence now.

2. Line60, “ horizontal variability at $O(10^0)$ to $O(10^3)$ m, which we term geoaoustic meso-scale variability”. In fact, $O(10^0)$ to $O(10^3)$ m is referred to as small size.

While the meteorological and oceanographic communities refer to different sizes as meso-scale, we specifically address seabed acoustics applications with this manuscript. For seabed acoustics problems, the range from 1 to 1000 m is considered meso-scale, since smaller resolution is only obtained through direct instruments (e.g., coring) while scales above that are most typical for acoustic ranging applications (e.g., submarine detection). Seabed properties between small and large scales significantly affect the acoustic signals, but have not been well studied yet due to the issues mentioned in the introduction. This manuscript presents a viable solution to some of those issues.

3.Line69-70, “Care must be taken with respect to position of source and array in the water column to ensure that direct and bottom-reflected paths can be separated”.How to realize it?

We have added the following sentence in line 69 (L. 69-70):
“This is done by keeping the source and receiver sufficiently far from the seafloor.”

4. Line79, “bulk density, sound speed, and sound attenuation (the latter two are frequency dependent)”. As we know, sound speed is independent of frequency.

The fact that sound speed is dependent on frequency (i.e., dispersion) is generally well known and accepted, although in many cases this dependence is neglected. It is generally recognized that sediment sound speed is frequency-dependent, e.g., the well-known Biot theory (Biot 1956, ref. below) and the Viscous Grain Shearing theory (ref. 26, 38, 39 (upd.: 26, 39, 40)) both predict a frequency-dependent sound speed. Dispersion in seabed sediments has been investigated and reported in ref. 28 and 35, and Belcourt et al. 2019 (ref. below), for a more direct measurement of the frequency dependence of saturated sediments see Williams et al., 2002 (ref. below). Furthermore, for an attenuating medium, in order to satisfy causality, the sound speed and attenuation frequency dependencies are linked by the Kramers-Kronigs relations. These say that sound speed is independent of frequency only if the attenuation goes as frequency to the second power over all frequencies, which is not the case for marine sediments. It is not uncommon practice, however, to ignore the frequency dependence, which may not be important in some problems.

We have added a sentence in Line 75 (L. 79-81) to emphasize the physical background:
“Note that the GS model satisfies causality which leads to frequency dependence of attenuation and compressional wave velocity.”

References:

- Biot, M. A. (1956). Theory of propagation of elastic waves in a fluid-saturated porous solid. I. Low frequency range. *The Journal of the Acoustical Society of America*, 28(2), 168-178.
- Biot, M. A. (1956). Theory of propagation of elastic waves in a fluid-saturated porous solid. II. Higher frequency range. *The Journal of the Acoustical Society of America*, 28(2), 179-191.
- Belcourt, J., Holland, C. W., Dosso, S. E., Dettmer, J., & Goff, J. A. (2019). Depth-dependent geoacoustic inferences with dispersion at the New England Mud Patch via reflection coefficient inversion. *IEEE Journal of Oceanic Engineering*, 45(1), 69-91.
- Williams, K. L., Jackson, D. R., Thorsos, E. I., Tang, D., & Schock, S. G. (2002). Comparison of sound speed and attenuation measured in a sandy sediment to predictions based on the Biot theory of porous media. *IEEE journal of oceanic engineering*, 27(3), 413-428.

5. Line78-79, “The GS parameters can be transformed to sediment parameters of bulk density, sound speed, and sound attenuation” should be provided by some references to further explain it.

We have added the relevant reference (26) to that sentence. (L. 83)

6.Line127, “Structures deeper than this denser layer have higher uncertainties” is difficult to be understood.

Thank you for noticing, we have replaced the unclear “this denser layer” with “the erosional unconformity”. (L. 132)

7. Line285, “The LFM pulses have good noise rejection characteristics; by match-filtering the data, all noise is reduced that exhibits different spectral or temporal characteristics than the LFM”. It is difficult to understand the sentence.

We have added a new reference to that sentence for the matched filter: (L. 292)

Buttkus, B. (2012). *Spectral analysis and filter theory in applied geophysics*. Springer Science & Business Media.